# Phyto-Computational Intervention of Diabetes Mellitus at Multiple Stages Using Isoeugenol from *Ocimum tenuiflorum*: A Combination of Pharmacokinetics and Molecular Modelling Approaches

**DOI:** 10.3390/molecules27196222

**Published:** 2022-09-22

**Authors:** Reshma Mary Martiz, Shashank M. Patil, Deepika Thirumalapura Hombegowda, Abdullah M. Shbeer, Taha Alqadi, Mohammed Al-Ghorbani, Ramith Ramu, Ashwini Prasad

**Affiliations:** 1Department of Biotechnology and Bioinformatics, School of Life Sciences, JSS Academy of Higher Education and Research, Mysore 570015, India; 2Department of Microbiology, School of Life Sciences, JSS Academy of Higher Education and Research, Mysore 570015, India; 3Department of Pharmacology, School of Life Sciences, JSS Academy of Higher Education and Research, Mysore 570015, India; 4Department of Surgery, Faculty of Medicine, Jazan University, Jazan 45142, Saudi Arabia; 5Department of Biology, Adham University College, Umm Al-Qura University, Makkah 21955, Saudi Arabia; 6Department of Chemistry, College of Science and Arts, Ulla, Taibah University, Madina 41477, Saudi Arabia; 7Department of Chemistry, College of Education, Thamar University, Thamar 425897, Yemen

**Keywords:** diabetes mellitus, *Ocimum tenuiflorum*, isoeugenol, in silico approach, molecular docking, molecular dynamics simulations, binding free energy calculations

## Abstract

In the present study, the anti-diabetic potential of *Ocimum tenuiflorum* was investigated using computational techniques for α-glucosidase, α-amylase, aldose reductase, and glycation at multiple stages. It aimed to elucidate the mechanism by which phytocompounds of *O. tenuiflorum* treat diabetes mellitus using concepts of druglikeness and pharmacokinetics, molecular docking simulations, molecular dynamics simulations, and binding free energy studies. Isoeugenol is a phenylpropene, propenyl-substituted guaiacol found in the essential oils of plants. During molecular docking modelling, isoeugenol was found to inhibit all the target enzymes, with a higher binding efficiency than standard drugs. Furthermore, molecular dynamic experiments revealed that isoeugenol was more stable in the binding pockets than the standard drugs used. Since our aim was to discover a single lead molecule with a higher binding efficiency and stability, isoeugenol was selected. In this context, our study stands in contrast to other computational studies that report on more than one compound, making it difficult to offer further analyses. To summarize, we recommend isoeugenol as a potential widely employed lead inhibitor of α-glucosidase, α-amylase, aldose reductase, and glycation based on the results of our in silico studies, therefore revealing a novel phytocompound for the effective treatment of hyperglycemia and diabetes mellitus.

## 1. Introduction 

Type-2 diabetes mellitus (T2DM) is a chronic metabolic disease characterized by hyperglycemia, in which the body’s metabolism is disrupted as a result of abnormalities in the insulin levels [1]. Prolonged hyperglycemic conditions lead to diabetes mellitus, which in turn results in the damage, dysfunction, and failure of various organs. Carbohydrate digestive enzymes, such as α-glucosidase and α-amylase, play a crucial role in fueling hyperglycemia by releasing monosaccharides in the course of digestion [2,3,4]. Therefore, the inhibition of carbohydrate digestive enzymes proves to be an essential part of treating diabetes mellitus. 

Furthermore, blood glucose levels that are too high cause a significant flow of glucose into the polyol pathway, where it is converted to sorbitol by aldose reductase [5]. Because the metabolism is impaired by the enzyme sorbitol dehydrogenase, sorbitol accumulates in the kidneys, nerves, and retina in diabetes patients. Additionally, sorbitol accumulation causes microvascular problems and a variety of cardiovascular diseases, which the aldose reductase enzyme can successfully prevent [6]. Thus, aldose reductase can also serve as an important target that can be inhibited in order to prevent sorbitol accumulation, which is associated with a number of microvascular and cardiovascular issues [7]. 

Furthermore, protein glycation in diabetes mellitus causes a partial impairment of activity as a result of prolonged hyperglycemia. The non-enzymatic nucleophilic addition reaction of the carbonyl residue of sugar with the free amino group of proteins forms a reversible Schiff base, which eventually gives rise to a more stable Amadori product. The Amadori products are then subjected to a series of dicarbonyl intermediate-mediated reactions, creating an unspecified class of compounds known as advanced glycation end products (AGEs). These AGEs accumulate in tissues and are the source of micro- and macro-vascular problems in diabetics [8,9]. It has been suggested that changes in lifestyle, such as increased physical activity and consumption of a diet rich in plant-derived foods (e.g., whole grains, fruits, and vegetables) might prevent 90% of T2DM cases [10]. Plant phytochemicals, such as polyphenolic compounds, as well as vitamins, minerals, and dietary fiber, have been linked to the health advantages of plant-derived products. Polyphenols have been demonstrated to lessen the severity of T2DM symptoms (such as fasting and postprandial hyperglycemia) by inhibiting disaccharidases (such as α-amylase and α-glucosidase) in the gut lumen [11].

*Ocimum tenuiflorum*, commonly known as tulsi, is a fragrant shrub of the basil family *Lamiaceae*, which is native to the eastern globe tropics, and is said to have originated in north-central India. In Ayurveda, it aids in the treatment of cough, asthma, diarrhea, fever, dysentery, arthritis, eye diseases, indigestion, gastric ailments, etc. The pharmacological advantages of tulsi have been demonstrated in numerous in vitro, animal, and human studies. *O. tenuiflorum* is therefore a valuable source of phytoconstituents that can be applied in pharmacotherapeutic procedures. Tulsi has been found to have a variety of pharmacological and phytochemical characteristics, including anti-diabetic properties [12,13]. In the current scenario, in silico pharmacology techniques are becoming an essential aspect of the drug development process. Meanwhile, in order to strategically plan our biological investigations, we sought to assess the anti-diabetic potential of *O. tenuiflorum* phytocompounds utilizing bioinformatics methods. Computational techniques, such as molecular docking, molecular dynamics modelling, and binding free energy calculations, have been proven to give accurate predictions in the field of drug development [14]. When compared to in vitro and in vivo studies, they save a substantial amount of time and money. As a result, we intended to virtually screen the *O. tenuiflorum* phytochemicals obtained from the Indian Medicinal Plants, Phytochemistry, and Therapeutics (IMPPAT) database as potential inhibitors of α–glucosidase, α–amylase, human aldose reductase, and human serum albumin proteins using molecular docking simulation, molecular dynamics (MD) simulation, binding free energy calculations, and pharmacokinetic analysis. The findings of this study suggest that the phytochemicals from *O. tenuiflorum* are potent and can act as multiple target inhibitors of T2DM; thus, they should be taken into consideration for further evaluation. Through this study, we aim to identify a single lead potential inhibitor of all the target enzymes used.

## 2. Material and Methods

### 2.1. Data Retrieval and ADMET Profiling

The datasets of active phytocompounds were searched using the Indian Medicinal Plants, Phytochemistry and Therapeutics database (https://cb.imsc.res.in/imppat/) (accessed on 10 July 2022) [15]. The phytocompounds were retrieved from NCBI PubChem database (https://pubchem.ncbi.nlm.nih.gov/) (accessed on 10 July 2022). Further, the compounds were screened based on their ADMET properties. To deduce the pharmacokinetic properties and their functions inside the body, the ADMET study was conducted based on previous works of the authors using ADMETlab 2.0 (https://admetmesh.scbdd.com/) (accessed on 10 July 2022) [16,17]. 

### 2.2. Molecular Docking Simulation

The 3D X-ray crystal structures of the target proteins required for the study, α-glucosidase, α-amylase, human serum albumin (HSA), and human aldose reductase (HAR), were retrieved from the RCSB Protein Data Bank (https://www.rcsb.org/) (accessed on 10 July 2022), and their PDB IDs are IDHK, 1AO6, and 1IEI, respectively. The protein sequence of *Saccharomyces cerevisiae* α-glucosidase MAL-32 obtained from UniProt (UniProt ID: P38158) was used to build a protein model using SWISS-MODEL. The model was created using the X-ray crystal structure of *S. cerevisiae* isomaltase (PDB ID: 3AXH), which showed a 72% identical and an 84% comparable sequence at a resolution of 1.8 Å. The model was evaluated and found to be stable in the authors’ previous works [7,18]. The pre-preparation of the proteins and ligands and virtual screening of compounds were performed based on Patil et al. (2021) [19]. The binding site was predicted according to the literature available on the RCSB PDB database. The grid box was placed on the binding pockets of the respective target proteins. The size of the grid box was maintained as constant, with different coordinates (Table 1). The molecular docking protocol was validated according to a previous study, where the same proteins (homology-built model of α-glucosidase, α-amylase, and HAR) were used for the in silico experiments [7]. In the case of HSA, the protocol was validated using the literature available on RCSB PDB database [20]. Concurrently, the 3D structures of the ligands were obtained from PubChem in SDF format and were later converted into PDBQT format using OpenBabel 2.3.1 [21,22]. Finally, the prepared protein and ligand compounds were docked using AutoDock Vina 1.1.2, along with their controls. For α-glucosidase and α-amylase, acarbose was considered as a control. Meanwhile, for human serum albumin (HSA) and human aldose reductase (HAR), aminoguanidine and quercetin were considered as controls, respectively. The selection of the control drugs was based on the previous works of the authors [7]. The virtual screening and interaction studies were performed using AutoDock Vina 1.1.2 and BIOVIA Discovery Studios Visualizer 2021, respectively, based on Kumar et al. (2021) [23].

### 2.3. Molecular Dynamics Simulation

Based on the interaction analysis of the compounds, the best docked conformation was selected for the dynamic investigation using GROMACS-2018.1, which is a biomolecular software package [24]. The molecular dynamics simulation study was conducted in order to understand the complexes’ stability, flexibility, and their conformational changes according to the time interval. Based on the work of Patil et al. (2021a) [18] and Patil et al. (2021b) [19], the simulation was performed for 100 ns. The simulation was carried out using a nanosecond scale, and the pdb2gmx program protein was assigned with the CHARMM36 force field to obtain the protein topology, whereas the SwissParam server (https://www.swissparam.ch/) (accessed on 12 July 2022) [25,26] was used to obtain the ligand topology. Furthermore, the system was solvated using a TIP3 water model with a 10 Å cubic box. An appropriate number of Na^+^ and Cl^−^ counter ions were added to neutralize the whole system, and the concentration of 0.15 M was added to maintain the salt concentration. By using the steepest descent algorithm, the energy minimization of 50,000 steps was performed on the system. Furthermore, the system was equilibrated in two phases, including the NVT and subsequent NPT ensemble (1000 ps each), with a 310 K temperature and 1 bar pressure [26,27]. The MD trajectories obtained were the root mean square deviation (RMSD), root mean square fluctuation (RMSF), radius of gyration (Rg), SASA (solvent accessible surface area), and the ligand hydrogen bonds. The MD trajectories were plotted and analyzed using XMGRACE, based on the previous studies by the authors [28,29]. 

### 2.4. Binding Free Energy Calculation

The binding free energy calculation of the complex was estimated using the mechanics/Poisson–Boltzmann surface area (MM–PBSA) approach, using the g_mmpbsa program, which is a GROMACS plugin. The quantitatively estimated of MM–PBSA was performed according to the study conducted by Martiz et al. (2022) [30]. The calculation was performed using the last 50 ns frames, which were extracted from the MD trajectory [18,19].

### 2.5. Druglikeness, Pharmacokinetics, and PASS Analysis of the Representative Compounds

Details related to the druglikeness and pharmacokinetics of the representative compounds (isoeugenol, acarbose, quercetin, and aminoguanidine) were retrieved from the previous analysis (virtual screening using molecular docking and ADMET profiling). In addition, the pharmacological activity prediction using the PASS online tool (http://www.way2drug.com/passonline/) (accessed on 15 July 2022) was performed on the representative compounds. The PASS server evaluates whether the provided chemical compound(s) can have a specific pharmacological effect [31]. The outcomes were numerical and classified into “Pa” and “Pi,” where “Pa” is symbolizes potential activity, while “Pi” indicates the potential inactivity of the given compound. The compounds that are considered acceptable for a particular pharmacological activity have comparatively greater Pa values than Pi values (Pa > Pi) [18,19]. In this study, parameters such as α-glucosidase inhibition, α-amylase inhibition, AGE-related disorder treatment, and HAR inhibition were assessed.

## 3. Results and Discussion

### 3.1. Virtual Screening through ADMET and Molecular Docking Simulation

Prior to performing the molecular docking, the in silico druglikeness and toxicity predictions were carried out in order to understand their biological activities and toxic effects. The screening results of all the phytocompounds of *O. tenuiflorum* are given in the Appendix A. 

Meanwhile, the ADMET screening results of the 26 selective compounds are given in Table 2. The predicted outcomes showed that most of the phytocompounds satisfy Lipinski’s rule of five, which is a commonly used criteria for classifying the compounds as drugs [32]. The oral bioavailability (OB) and blood–brain barrier (BBB) showed a better permeation. OB is one of the most significant pharmacokinetic features in addition ADME properties. Whereas, in the case of the TPSA, the compounds with <140 Å TPSA value were considered as more flexible and could interact better with the target protein [33]. As evident from Table 2, the values of the selected properties were well within range, and the molecules showed excellent percentages of human oral absorption.

After the ADMET screening, the docking study was carried out for the 26 selected phytocompounds. Table 3 displays the docking results of the compounds with α-glucosidase, α-amylase, HSA, and HAR as their target proteins. From Table 3, it can be concluded that all of the molecules have significantly lower docking scores (the more negative the docking score is, the better the binding is). Out of all the phytocompounds docked, isoeugenol was selected as a single multi-protein inhibitor based on its pharmacokinetic properties and binding efficiency, since our aim was to discover this type of inhibitor. 

In this regard, all the pharmacokinetic parameters, the binding affinity, total number of intermolecular interactions, and total number of hydrogen bonds were taken into consideration. Isoeugenol had the highest docking score in comparison to the other docked complexes and in the case of all the protein targets. The π-π stacking interactions, halogen bonding, hydrogen bonding, and aromatic hydrogen bonding were the typical interactions observed. Further analysis, in order to understand the differences in the docking scores, was carried out using MD simulations.

In the case of α-glucosidase, isoeugenol had the better binding affinity and was bound within the inhibitor binding site of the protein. In comparison with the control, acarbose, isoeugenol formed a higher number of bonds, as presented in Figure 1. The isoeugenol complex formed a total of nine intermolecular interactions, which included four hydrogen bonds with Asp68, Arg439, Glu276, and Asp214. A single electrostatic bond was formed with Asp349. Hydrophobic π-π stacked bond bounds were formed via Phe177, whereas Tyr71, Phe157, and Phe177 formed π-alkyl with the ligand. The predicted binding interaction results are in accordance with the previous works [17,21,34]. Meanwhile, the control compound, acarbose, formed a total of seven intermolecular bonds, which is less than the isoeugenol compound, of which six were hydrogen bonds formed via Asn241, Arg439, Asp408, Pro309, and His239. A hydrophobic π-sigma bond was formed between acarbose and His279, whereas Thr307 and Asp349 were found to have formed an unfavorable acceptor–acceptor bond. The visualization of the binding interaction of isoeugenol and acarbose with α-glucosidase is given in Figure 1.

Based on the α-amylase-bound isoeugenol and acarbose docking study, isoeugenol was predicted to bind within the inhibitory binding site. A total of seven intermolecular bonds were predicted, of which four were hydrogen bonds via Arg398, Thr11, and Asp402. It also formed hydrophobic pi-pi-shaped bonds via Phe335 and pi-alkyl via Pro4 and Arg398. Meanwhile, a total four hydrogen bonds were formed between the protein and acarbose, and one unfavorable bond was formed with His331. The docking results were found to be in accordance with the previous studies [23,29,35]. The binding interactions of isoeugenol and acarbose with α-amylase are visualized in Figure 2. 

Meanwhile, in the case of HAR, both isoeugenol and quercetin were bound within the inhibitory pocket of protein. Based on the predicted complex, isoeugenol formed a total of 13 bonds, of which Cys298 formed a hydrogen bond. The hydrophobic pi-sigma and pi-pi stacked bonds were formed via Trp111. The alkyl and pi-alkyl bonds were formed with Val47, Cys80, Leu300, Trp20, Trp79, Trp111, and Phe122. In comparison, quercetin had eight intermolecular interactions, of which one hydrogen bond and two donor–donor and acceptor–acceptor unfavorable bonds were formed via Tyr309 and Cys298, thus indicating that isoeugenol might form a better stable complex than quercetin. The docking results were found to be in accordance with the previous studies [7]. The binding interactions of isoeugenol and quercetin with HAR are visualized in Figure 3. 

In the case of HSA, the isoeugenol compounds were predicted to result in more non-bonded interactions (8) when compared to aminoguanidine (7). A total of eight inter-molecular interactions were predicted, of which Glu354 and Arg209 formed hydrogen bonds, while hydrophobic alkyl and pi-alkyl were formed with Lys212, Val216, Lys351, Ala213, Leu327, and Ala350. These residues are present in the vicinity of fatty acid site 4 (FA4), which accommodates the methylene tails of lipids bound to this site. In addition, the ligands occupied the same binding site as the co-crystallized inhibitor ligand ibuprofen, in accordance with the previous study [20]. According to this study, binding in the polar patch of the binding pocket induces the conformational changes in the protein. In comparison, aminoguanidine formed an unfavorable donor–donor bond between the ligand and Arg197. The binding interactions of isoeugenol and quercetin with HSA are visualized in Figure 4.

Thus, based on the overall study, using different target proteins, the isoeugenol compound was predicted to have a better stability during complex formation, and the compound was identified as binding within the inhibitory binding pocket, suggesting that it might act as an inhibitor drug. The results obtained were in accordance with the previous studies [36,37].

### 3.2. Molecular Dynamics Simulation

Molecular dynamics simulation was performed to provide insight into the protein–ligand stability and protein structural flexibility of the docked complexes. The simulations of isoeugenol, along with the respective controls (acarbose for α-glucosidase and α-amylase, aminoguanidine for HSA, and quercetin for HAR), which bound to the targets α –glucosidase, α-amylase, HSA, and HAR, respectively, were carried out using the docked structure as a starting geometry [38]. Figure 5 represents the plot of the trajectories of the isoeugenol and acarbose complexes bound to α-glucosidase, along with apo-protein. Throughout the simulation, the RMSD values of the complexes of isoeugenol and acarbose, as well as protein α-glucosidase, show periodic variations, and isoeugenol was found within the inhibitor binding site. The isoeugenol complex was found to be stable after 80 ns, whereas fluctuation was found throughout the simulation in the case of the acarbose complex. The RMSF plot was analyzed to discern each residue’s fluctuations during the period of the simulation (100 ns). Both the complexes that bound to protein were on par, with almost similar pattern. The protein model was shown to be relatively stable at both the N- and T-terminals. The isoeugenol complex showed lower fluctuations, indicating that its interaction may be superior. The radius of gyration (Rg) of the complexes and apo-protein was determined, since it represents the structural compactness of the structure. The Rg and SASA values showed similar patterns throughout the experiment, with no fluctuations. Based on the hydrogen bond analysis, it can be predicted that structural re-agreement may have occurred during the simulation, as the number of hydrogen bonds increased when compared to the docking process; thus, it can be predicted that the isoeugenol complex has better stability compared to the acarbose complex. The simulation results complemented those of recent works that used the same protein model of α-glucosidase [18,23]. The MD simulation analysis of isoeugenol and acarbose demonstrated that both the complexes are found within the inhibitor binding site and formed persistent contacts, which may contribute to the stability of the complexes (Figure 5). Table 4 depicts both isoeugenol and acarbose complexed with α-glucosidase and the MD trajectory values.

In the case of the α-amylase-bound complexes, the RMSD plot indicates that the isoeugenol complex is bound within the inhibitor binding site, whereas a much higher deviation can be seen in the case of the acarbose complex, which might indicate that isoeugenol is more stable than the acarbose complex. Both complexes and the apo-protein showed more or less identical oscillation patterns in the RMSF evaluation. The higher fluctuation can be seen in the loop region of the structures that were studied, which indicates that there might be a chance of high mobility. Meanwhile, compared to the isoeugenol complex, both the acarbose complex and apo-protein showed high fluctuation, which may suggest that the instability with the structure. To understand the structure compactness and the stability of the complexes formed, both Rg and SASA were evaluated. Based on the evaluation, it was observed that the Rg values of both the complexes, isoeugenol and acarbose, showed similar pattern. Thus, it can be said that the complexes were compact throughout the simulation. Furthermore, to understand if any structural rearrangement occurred within the complexes, the ligand H-bond was analyzed. Based on the H-bond plot analysis, he acarbose complex showed the same number of hydrogen bonds as the isoeugenol complex, which was in accordance with our previous study. The outcomes of the MD simulation of α-amylase were found to be in accordance with the previous studies [21,26]. The graphical representation of the MD simulation plot is shown in Figure 6, and the trajectory values are given in Table 5.

The MD simulation plot analysis of the HAR-bound isoeugenol and quercetin complexes is shown in Figure 7. The RMSD plot illustrates that both the complexes are bound to the protein within the inhibitory site. Based on the plot, it can be said that isoeugenol bound to HAR stabilized after 20 ns, whereas quercetin showed a slight variation throughout the simulation. Thus, based on the RMSD evaluation, it may be said that the isoeugenol complex is more stable compared to quercetin complex. The simulation result is in accordance with the authors’ previous studies [7,39]. The RMSF plot shows high fluctuations between the residues, and both the complexes, as well as the apo-protein, showed similar patterns throughout the simulation. The Rg values of both the apo-protein and isoeugenol complex show a similar pattern, which might indicate the better compactness of the structure, whereas, based on the SASA plot, it can be observed that all the structures, the apo-protein, isoeugenol complex, and quercetin complex, yielded values that are on par and show a rather similar pattern. Finally, based on the H-bond analysis, it can be seen that the isoeugenol bound complex has a maximum of seven hydrogen bonds, compared to the quercetin docked complex, which indicates that the structure may have undergone structural rearrangement (Figure 7). The trajectory values of the MD simulation are given in Table 6.

In the case of HSA, the RMSD plot analysis showed that both the complexes and apo-protein showed rather similar patterns of variation throughout the MD simulation. Both the complexes are bound within the inhibitory site of the protein. Similar high fluctuations were seen at the terminal region of the RMSF plot, which indicates that there might be a chance of high mobility between the residues (480–590). However, based on the RMSF plot, overall, the fluctuations of the protein-aminoguanidine complex were found to be greater. The Rg value was evaluated, indicating the compactness of the structure during the complex formation, and based on the plot, it can be observed that the isoeugenol complex may have a better compactness compared with the aminoguanidine complex. A similar pattern can be seen even in the SASA plot. Finally, based on H-bond analysis, it can be predicted that structural rearrangement might have taken place. The results of the MD simulation for HSA were found to be in accordance with the previous studies [40,41]. The graphical visualization of the MD trajectory plot and values are given in Figure 8 and Table 7, respectively.

### 3.3. Binding Free Energy Calculations

Based on the free binding energy calculations, it can be predicted that van der Waal’s energy and the binding energies had substantial impacts on the complex formation. Based on the energy calculation, the predicted results were, mostly, energetically viable. According to the predicted results, isoeugenol bound to the α-glucosidase complex (−224.811 kJ/mol) showed the highest binding free energy when compared with all the other complexes (Table 8). Van der Waal’s energy and binding free energy were shown to be the primary contributors to the formation of the complexes when compared to the other energies. Furthermore, when compared to the protein-control complexes, the protein-isoeugenol complexes were found to have higher (more negative) binding free energies, which indicates that the protein-control complexes have a weaker interaction and binding affinity compared to the protein-isoeugenol complexes. This study result revealed a similar pattern to that observed in previous studies that performed binding free energy calculations for α-glucosidase, α-amylase, HAR [7], and HSA [42]. The predicted values of the energies calculated are summarized in Table 8 and were obtained using the MMPBSA technique.

### 3.4. Druglikeliness, Pharmacokinetic, and PASS Analysis of the Representative Compounds

In terms of the druglikeness properties, all the molecules except acarbose were found to be in accordance with Lipinski’s rule of five. During the analysis of the pharmacokinetic properties, acarbose and quercetin were found to violate the oral bioavailability parameter. However, studies have shown that both acarbose [43,44] and quercetin [45] possess minimal bioavailability. This makes the drugs effectively unsuitable for oral consumption. moreover, both of these compounds were predicted to cause liver injuries. This arises due to their toxicity and carcinogenic properties [46,47]. Therefore, ADMET profiling of the compounds revealed that both acarbose and quercetin are unfavorable for oral consumption. However, isoeugenol was predicted to be successful in all the investigations conducted. These reports indicate that isoeugenol may be used in in vitro and in vivo investigations. However, phytocompounds could be used as alternative therapeutics due to their minimal adverse effects on the human metabolism [48,49,50]. Table 9 depicts the druglikeness and pharmacokinetic properties of the representative compounds, whereas Figure 9 shows their pharmacokinetic mapping. The radar diagrams demonstrate that all the compounds were found to be within the acceptable boundary regarding the druglikeness properties, except for acarbose.

In addition, PASS pharmacological action predictions were also conducted to examine properties including α-glucosidase inhibition, α-amylase inhibition, AGE-related disorder treatment, and HAR inhibition. Isoeugenol was found to have more ‘Pa’ values than ‘Pi’, which indicates its positive activity in regard to all the parameters. However, acarbose was not predicted to exhibit a pharmacological action for the treatment of AGE-related disorders. In addition, aminoguanidine was found to be inactive in regard to all the parameters investigated. Moreover, quercetin was found to be inactive in terms of α-amylase inhibition and the treatment of AGE-related disorders. These results indicate that isoeugenol may act as a potential lead compound, a hypothesis which requires thorough investigation using in vitro and animal models. The results from the PASS analysis of the representative compounds are depicted in Table 10.

## 4. Conclusions

Plant-based antidiabetic drug development has been a stumbling block, with few promising results, as most of the drugs have yet to pass the stage of clinical trials. Conventional chemotherapeutics have been widely used in the pharmaceutical industry due to their rapid action, mechanism, and economic viability. In the current investigation, we carried out a combination of in silico investigations of the phytocompounds of *O. tenuiflorum* and proposed isoeugenol as a potential inhibitor of α-glucosidase, α-amylase, human serum albumin, and human aldose reductase. Isoeugenol showed the highest probability of all the phytocompounds to act as a multi-target inhibitor of all the target enzymes mentioned above. This can reduce the biological enzymatic activity of the target enzymes, which could bring about a decline in hyperglycemia. The MD simulations and binding free energy calculations, through which the binding of isoeugenol with the drug targets was validated, supported the docking results. To summarize, isoeugenol has the potential to act as a multi-target inhibitor that can be used to treat diabetes mellitus at various stages of the disorder. In the near future, isoeugenol could be assessed using in vitro, in vivo, and then clinical trials in order to discover its potential as an antidiabetic drug targeting different stages of diabetes mellitus.

## Figures and Tables

**Figure 1 molecules-27-06222-f001:**
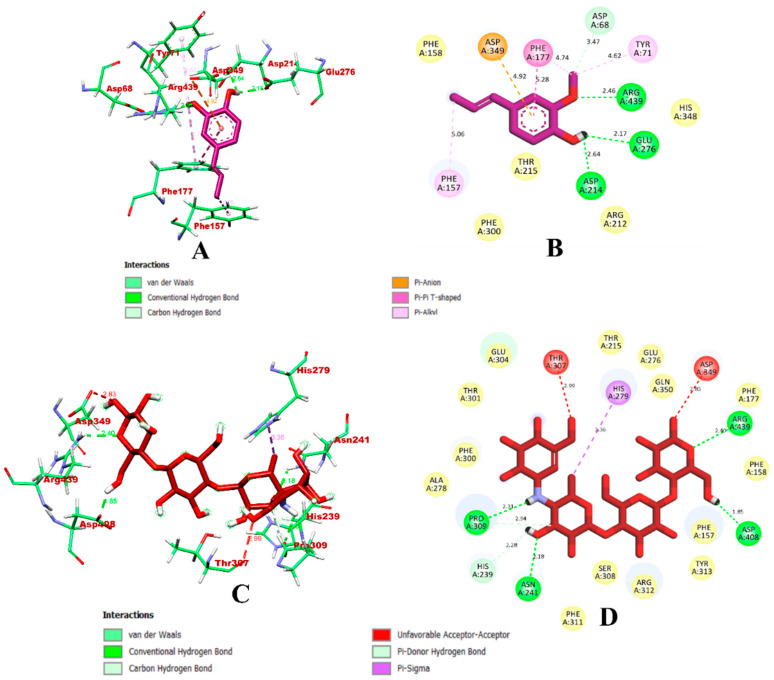
Visualization of the docking simulation of the compounds with α-glucosidase. (**A**,**B**) Interaction of isoeugenol (purple) visualized in 3D and 2D, respectively. (**C**,**D**) Interaction of acarbose (red) visualized in 3D and 2D, respectively.

**Figure 2 molecules-27-06222-f002:**
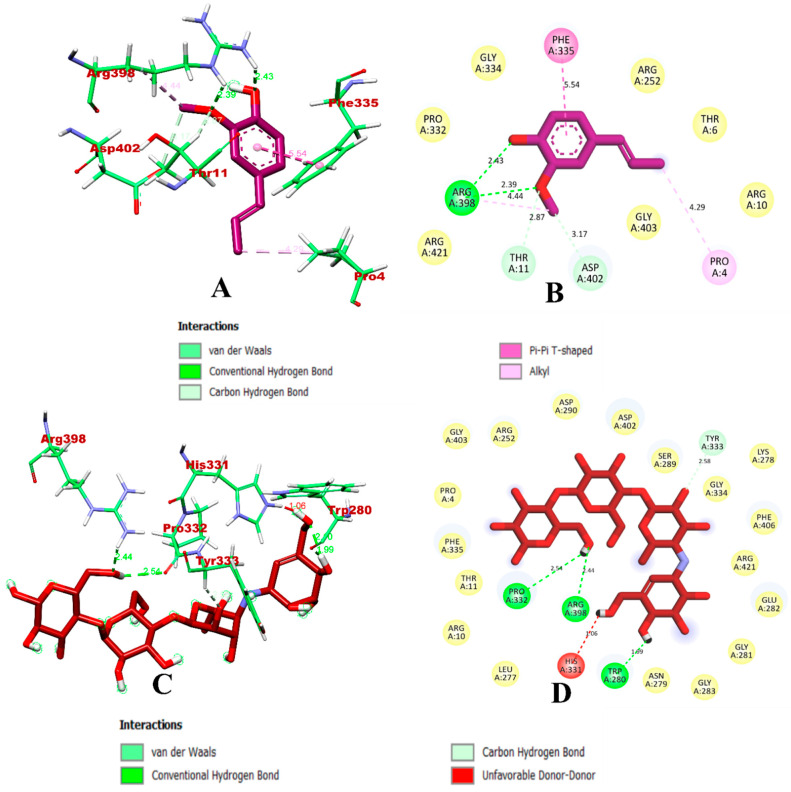
Visualization of the docking simulation of compounds with α-amylase. (**A**,**B**) Interaction of isoeugenol (purple) visualized in 3D and 2D, respectively. (**C**,**D**) Interaction of acarbose (red) visualized in 3D and 2D, respectively.

**Figure 3 molecules-27-06222-f003:**
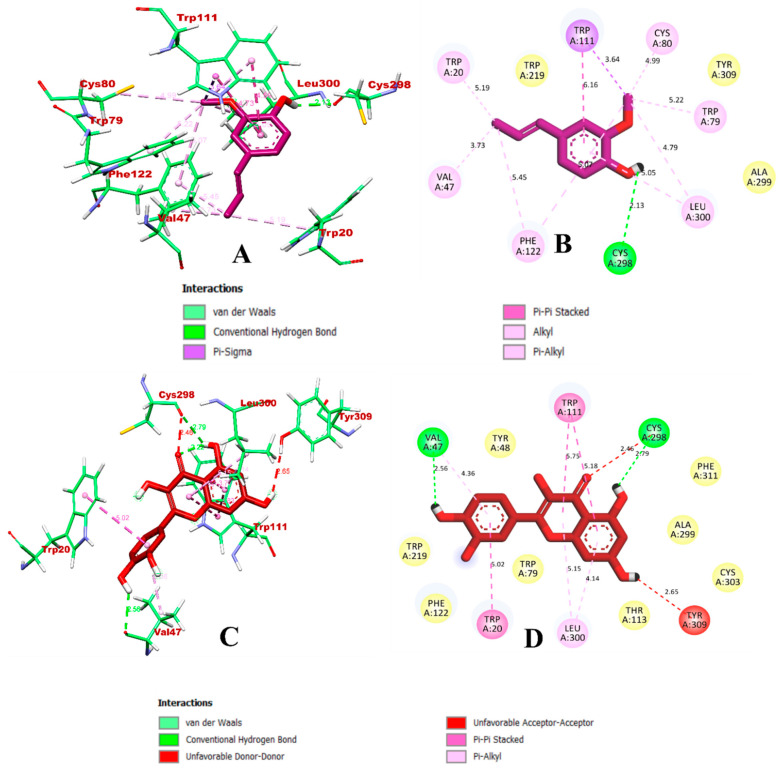
Visualization of the docking simulation of compounds with HAR. (**A**,**B**) Interaction of isoeugenol (purple) visualized in 3D and 2D, respectively. (**C**,**D**) Interaction of quercetin (red) visualized in 3D and 2D, respectively.

**Figure 4 molecules-27-06222-f004:**
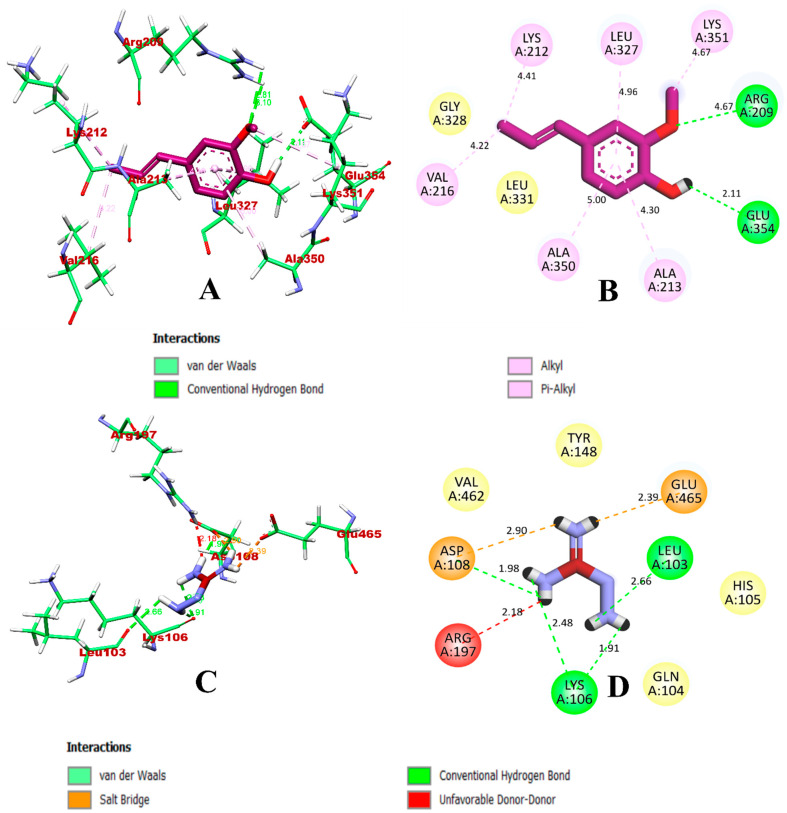
Visualization of the docking simulation of compounds with HSA. (**A**,**B**) Interaction of isoeugenol (purple) visualized in 3D and 2D, respectively. (**C**,**D**) Interaction of aminoguanidine (red) visualized in 3D and 2D, respectively.

**Figure 5 molecules-27-06222-f005:**
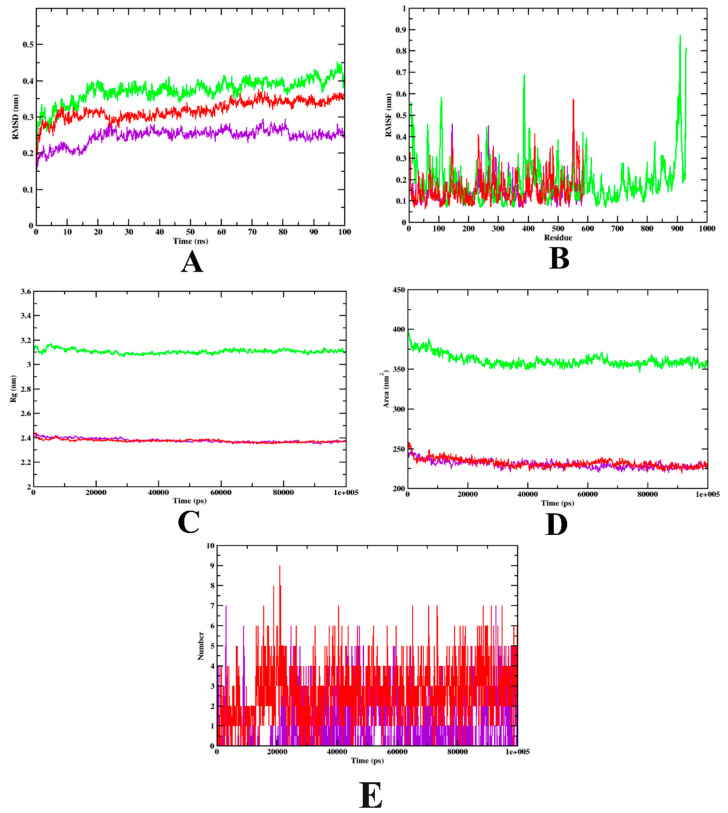
Analysis of RMSD, RMSF, Rg, SASA, and the number of hydrogen bonds of isoeugenol- (purple) and acarbose-bound (red) α-glucosidase complexes, as well as apo-protein α-glucosidase (green), at 100 ns. (**A**) Time evolution of RMSD values of both the complexes along with the protein. (**B**) RMSF. (**C**) Radius of gyration (Rg). (**D**) SASA. (**E**) Hydrogen bonds.

**Figure 6 molecules-27-06222-f006:**
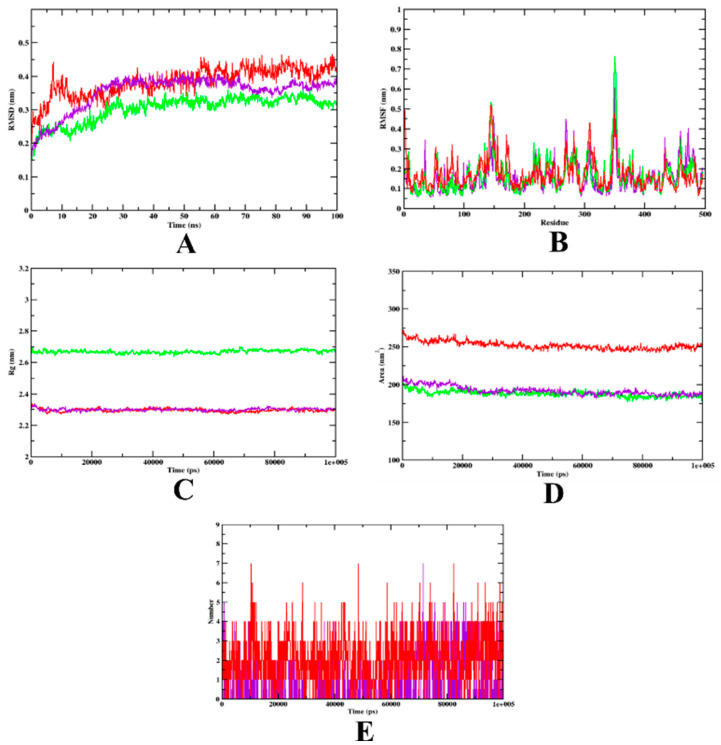
Analysis of RMSD, RMSF, Rg, SASA, and the number of hydrogen bonds of isoeugenol- (purple) and acarbose-bound (red) α-amylase complexes, as well as apo-protein α-amylase (green), at 100 ns. (**A**) Time evolution of RMSD values of both the complexes along with the protein. (**B**) RMSF. (**C**) Radius of gyration (Rg). (**D**) SASA. (**E**) Hydrogen bonds.

**Figure 7 molecules-27-06222-f007:**
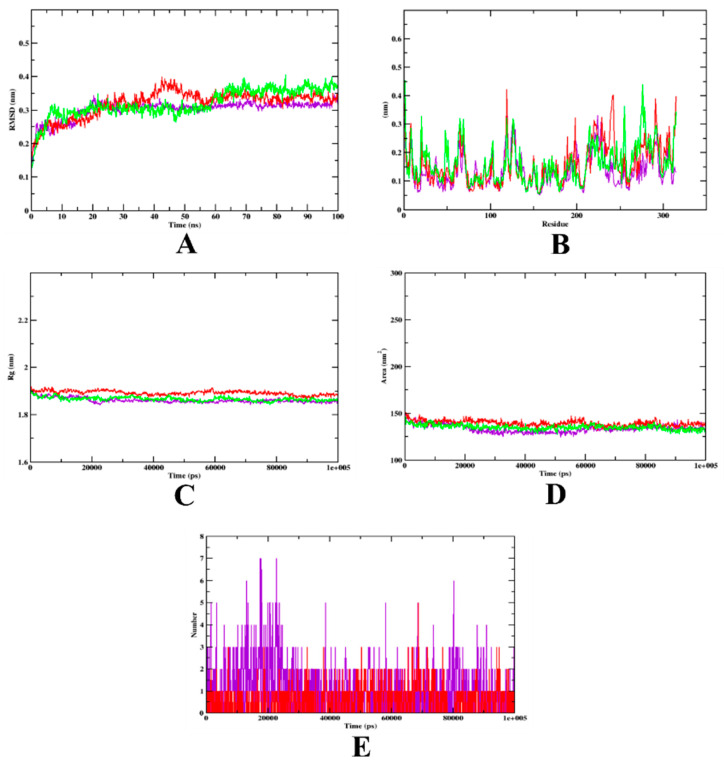
Analysis of RMSD, RMSF, Rg, SASA, and the number of hydrogen bonds of isoeugenol- (purple) and quercetin- bound (red) HAR complexes, as well as apo-protein α-glucosidase (green), at 100 ns. (**A**) Time evolution of RMSD values of both the complexes, along with the protein. (**B**) RMSF. (**C**) Radius of gyration (Rg). (**D**) SASA. (**E**) Hydrogen bonds.

**Figure 8 molecules-27-06222-f008:**
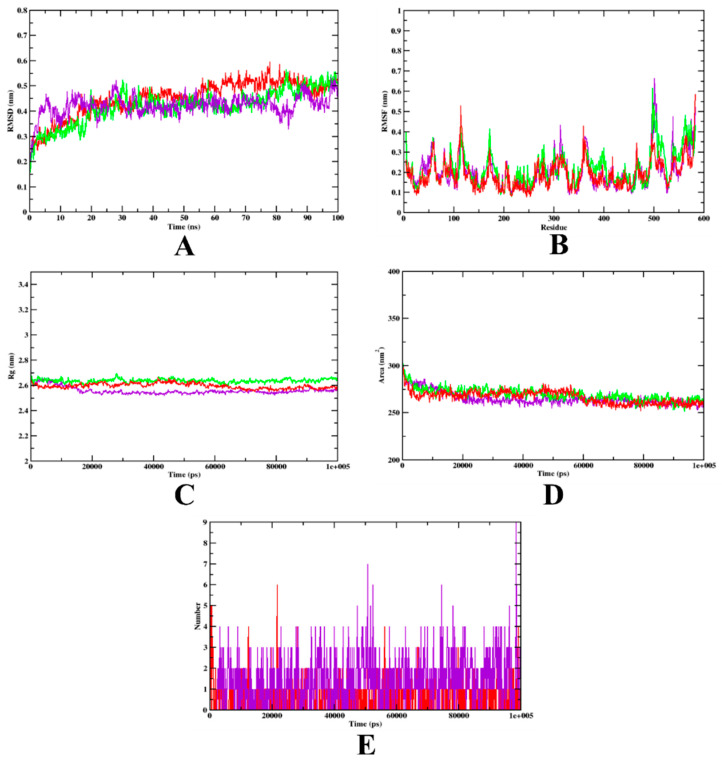
Analysis of RMSD, RMSF, Rg, SASA, and the number of hydrogen bonds of isoeugenol- (purple) and aminoguanidine-bound (red) HSA complex, as well as apo-protein HSA (green), at 100 ns. (**A**) Time evolution of RMSD values of both the complexes, along with the protein. (**B**) RMSF. (**C**) Radius of gyration (Rg). (**D**) SASA. (**E**) Hydrogen bonds.

**Figure 9 molecules-27-06222-f009:**
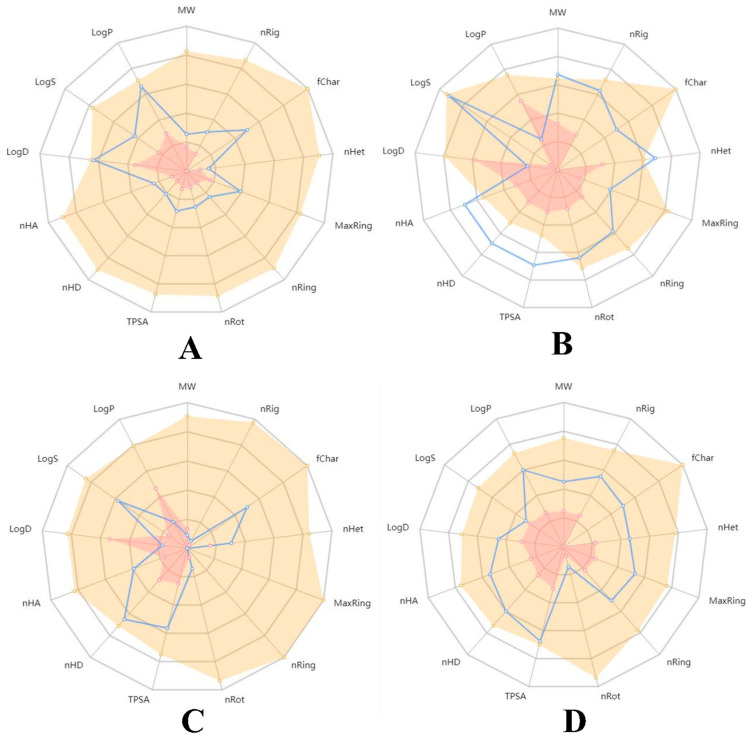
Pharmacokinetic mapping of the representative compounds: (**A**) isoeugenol, (**B**) acarbose, (**C**) aminoguanidine, and (**D**) quercetin.

**Table 1 molecules-27-06222-t001:** The position of the grid box placed on binding pockets of the respective target proteins.

Enzyme Targets	Coordinates of the Grid Box			Size of the Grid Box
	**x**	**y**	**z**	
α-glucosidase	−17.48 Å	−8.62 Å	−19.65 Å	40 Å × 40 Å × 40 Å
α-amylase	103.46 Å	37.17 Å	19.60 Å	40 Å × 40 Å × 40 Å
HSA	8.24 Å	2.58 Å	−14.75 Å	40 Å × 40 Å × 40 Å
HAR	−5.06 Å	0.19 Å	9.94 Å	40 Å × 40 Å × 40 Å

**Table 2 molecules-27-06222-t002:** Pharmacokinetics and ADMET screening results of selective compounds of *O. tenuiflorum* obtained from the ADMETlab 2.0 server.

Sl. No	Compound Names	OB (OB ≥ 30%)	BBB	DHL (HL < 3 h)	LR5	IEP (Caco-2 Cells)	DILI	CL (CL > 15 mL/min/kg)	MW (100~600)	HBA (0~12)	HBD (0~7)	TPSA (0~140)	PAINS
1	(−)-Alloaromadendrene	Pass	Pass	0.671	Accepted	−4.711	Negative	5.356	212.150	4	2	58.200	0
2	(−)-Camphene	Pass	Pass	0.013	Accepted	−4.756	Negative	16.686	414.390	1	1	20.230	0
3	(−)-Linalool	Pass	Pass	0.040	Accepted	−4.577	Negative	13.563	204.190	0	0	0.000	0
4	(+)-α-Phellandrene	Pass	Pass	0.077	Accepted	−4.463	Negative	9.346	136.130	0	0	0.000	0
5	(+)-Endo-β-bergamotene	Pass	Pass	0.493	Accepted	−4.375	Negative	9.738	154.140	1	1	20.230	0
6	(1S)-1,7,7-Trimethylbicyclo[2.2.1]heptan-2-one	Pass	Pass	0.617	Accepted	−4.383	Negative	12.660	136.130	0	0	0.000	0
7	(1S,2R,4S)-(−)-Bornyl acetate	Pass	Pass	0.063	Accepted	−4.466	Negative	16.946	204.190	0	0	0.000	0
8	(E)-β-ocimene	Pass	Pass	0.243	Accepted	−4.552	Moderate	6.063	196.150	2	0	26.300	0
9	1S-α-Pinene	Pass	Pass	0.701	Accepted	−4.582	Negative	13.808	152.120	1	0	17.070	0
10	2,3-Dimethylaniline	Pass	Pass	0.678	Accepted	−4.434	Negative	14.171	136.130	0	0	0.000	0
11	3-Carene	Pass	Pass	0.114	Accepted	−4.303	Negative	15.022	136.130	0	0	0.000	0
12	4-Terpineol	Pass	Pass	0.583	Accepted	−4.255	Negative	10.496	121.090	1	2	26.020	0
13	Acetyleugenol	Pass	Pass	0.132	Accepted	−4.307	Negative	16.061	136.130	0	0	0.000	0
14	α-Fenchene	Pass	Pass	0.447	Accepted	−4.217	Negative	14.345	154.140	1	1	20.230	0
15	α-Terpineol	Pass	Pass	0.843	Accepted	−4.453	Moderate	8.457	206.090	3	0	35.530	0
16	β-caryophyllene	Pass	Pass	0.099	Accepted	−4.460	Negative	10.559	136.130	0	0	0.000	0
17	β-Pinene	Pass	Pass	0.527	Accepted	−4.193	Negative	8.942	154.140	1	1	20.230	0
18	*Cis*-Anethole	Pass	Pass	0.048	Accepted	−4.517	Negative	9.943	204.190	0	0	0.000	0
19	Cyclo(L-Val-L-Leu)	Pass	Pass	0.107	Accepted	−4.460	Negative	10.097	136.130	0	0	0.000	0
20	Dehydro-p-cymene	Pass	Moderate	0.638	Accepted	−4.440	Negative	11.146	148.090	1	0	9.230	0
21	Eucalyptol	Pass	Pass	0.568	Accepted	−4.344	Moderate	10.755	132.090	0	0	0.000	0
22	γ-Selinene	Pass	Pass	0.352	Accepted	−4.414	Negative	8.066	154.140	1	0	9.230	0
23	Geranyl acetate	Pass	Pass	0.088	Accepted	−4.577	Negative	13.350	204.190	0	0	0.000	0
24	Isoeugenol	Pass	Pass	0.506	Accepted	−4.420	Moderate	9.707	196.150	2	0	26.300	0
25	Myrcene	Pass	Moderate	0.880	Accepted	−4.579	Negative	13.435	164.080	2	1	29.460	0
26	Phytosterols	Pass	Pass	0.453	Accepted	−4.402	Moderate	13.108	136.130	0	0	0.000	0
27	Acarbose	Fail	Fail	0.546	Rejected	−6.149	Positive	0.373	645.250	19	14	321.170	0
28	Aminoguanidine	Pass	Moderate	0.714	Accepted	5.448	Negative	5.857	74.060	4	6	87.920	0
29	Quercetin	Fail	Pass	0.929	Accepted	−5.204	Positive	8.284	302.040	7	8	131.360	1

Note: OB: oral bioavailability, BBB: blood–brain barrier, DHL: drug half-life, LR5: Lipinski’s rule of five, IEP: intestinal epithelial permeability (Caco-2 cells), DILI: drug-induced liver injury, CL: clearness, MW: molecular weight, HBA: hydrogen bond acceptor, HBD: hydrogen bond donor, TPSA: topological polar surface area.

**Table 3 molecules-27-06222-t003:** Name of the compounds, respective binding affinity, and total non-bonded interactions and hydrogen bonds, with their respective target proteins.

Sl. No.	Compound Names	Binding Affinity (kcal/mol)				Total No. of Intermolecular Interactions				Total No. of Hydrogen Bonds			
		AG	AM	HSA	HAR	AG	AM	HSA	HAR	AG	AM	HSA	HAR
1	(−)-Alloaromadendrene	−7.3	−6.8	−6.3	−5.1	2	7	5	8	-	-	-	-
2	(−)-Camphene	−5.3	−5.4	−5.7	−6.2	3	3	3	7	-	-	-	-
3	(−)-Linalool	−5.8	−4.9	−5.2	−6.2	10	7	6	9	3	-	-	-
4	(+)-α-Phellandrene	−5.8	−5.6	−5.9	−6.5	5	4	6	7	-	-	-	-
5	(+)-Endo-β-bergamotene	−7.3	−6.1	−6.2	−7.3	6	5	6	7	-	-	-	-
6	(1S)-1,7,7-Trimethylbicyclo[2.2.1]heptan-2-one	−5.9	−5.5	−5.7	−6.3	2	4	3	3	-	2	-	-
7	(1S,2R,4S)-(−)-Bornyl acetate	−6.9	−5.7	−6.4	−6.4	2	3	3	3	1	-	1	2
8	(E)-β-ocimene	−5.7	−5.0	−5.8	−6.4	8	6	6	11	-	-	-	-
9	1S-α-Pinene	−5.5	−5.4	−6.2	−6.1	4	9	6	9	-	-	-	-
10	2,3-Dimethylaniline	−5.1	−5.3	−5.6	−5.9	3	5	7	3	3	3	1	-
11	3-Carene	−5.4	−5.5	−6.0	−6.5	3	4	6	6	-	-	-	-
12	4-Terpineol	−5.8	−5.7	−6.2	−6.2	6	3	5	7	1	-	-	-
13	Acetyleugenol	−6.3	−5.7	−6.5	−6.6	7	4	8	6	2	1	2	1
14	α-Fenchene	−5.6	−5.3	−5.8	−6.1	4	5	4	8	-	-	-	-
15	α-Terpineol	−6.2	−6.0	−6.2	−6.5	5	5	5	7	2	-	-	-
16	β-caryophyllene	−7.3	−6.0	−6.1	−7.0	1	1	2	3	-	-	-	-
17	β-Pinene	−5.4	−5.6	−6.3	−6.1	3	5	6	8	-	-	-	-
18	*Cis*-Anethole	−5.7	−5.4	−6.3	−5.9	7	4	6	6	-	-	-	-
19	Cyclo(L-Val-L-Leu)	−6.6	−5.6	−6.4	−6.3	4	3	1	1	-	-	-	-
20	Dehydro-p-cymene	−6.0	−5.8	−6.1	−6.8	5	6	7	10	-	-	-	-
21	Eucalyptol	−5.5	−5.3	−6.2	−6.2	1	5	4	7	1	-	2	-
22	γ-Selinene	−7.3	−6.7	−6.8	−7.4	1	4	5	8	-	-	1	1
23	Geranyl acetate	−6.3	−5.3	−6.3	−7.0	7	7	5	7	2	2	2	-
24	Isoeugenol	−7.6	−6.9	−6.8	−7.4	9	7	8	11	4	4	2	1
25	Myrcene	−5.6	−4.9	−5.9	−6.2	8	6	9	9	-	-	-	-
26	Phytosterols	−7.2	−6.6	−6.8	−4.0	6	7	3	2	-	-	-	-
27	Acarbose	−8.2	−7.4	-	-	7	4	-	-	6	4	-	-
28	Aminoguanidine	-	-	−8.0	-	-	-	7	-	-	-	3	-
29	Quercetin	-	-	-	−7.8	-	-	-	8	-	-	-	1

Note: AG: α-glucosidase, AM: α-amylase, HSA: human serum albumin, HAR: human aldose reductase.

**Table 4 molecules-27-06222-t004:** MD trajectory values of isoeugenol and acarbose complexed with α-glucosidase.

MD Trajectory Values	Apo-Protein	Protein-Acarbose Complex	Protein-Isoeugenol Complex
RMSD	0.30–0.40 nm	0.25–0.32 nm	0.20–0.25 nm
Rg	3.10–3.14 nm	2.39–2.45 nm	2.39–2.45 nm
SASA	350–370 nm^2^	240–250 nm^2^	240–250 nm^2^
Ligand H-bonds	-	9	7

**Table 5 molecules-27-06222-t005:** MD trajectory values of isoeugenol and acarbose complexed with α-amylase.

MD Trajectory Values	Apo-Protein	Protein-Acarbose Complex	Protein-Isoeugenol Complex
RMSD	0.20–0.30 nm	0.25–0.32 nm	0.20–0.25 nm
Rg	3.10–3.14 nm	2.39–2.45 nm	2.39–2.45 nm
SASA	350–370 nm^2^	240–250 nm^2^	240–250 nm^2^
Ligand H-bonds	-	7	7

**Table 6 molecules-27-06222-t006:** MD trajectory values of isoeugenol and quercetin complexed with HAR.

MD Trajectory Values	Apo-Protein	Protein-Quercetin Complex	Protein-Isoeugenol Complex
RMSD	0.20–0.36 nm	0.20–0.35 nm	0.20–0.30 nm
Rg	1.70–1.90 nm	1.90–1.92 nm	1.70–1.90 nm
SASA	140–150 nm^2^	140–150 nm^2^	140–150 nm^2^
Ligand H-bonds	-	5	7

**Table 7 molecules-27-06222-t007:** MD trajectory values of isoeugenol and aminoguanidine complexed with HSA.

MD Trajectory Values	Apo-Protein	Protein-Aminoguanidine Complex	Protein-Isoeugenol Complex
RMSD	0.25–0.45 nm	0.25–0.45 nm	0.25–0.45 nm
Rg	2.60–2.61 nm	2.60–2.61 nm	2.58–2.60 nm
SASA	270–280 nm^2^	270–280 nm^2^	270–280 nm^2^
Ligand H-bonds	-	6	7

**Table 8 molecules-27-06222-t008:** Binding free energy values of the target proteins complexed with ligands.

Protein-Ligand Complexes	Types of Binding Free Energies				
	Van Der Waal’s Energy (kJ/mol)	Electrostatic Energy (kJ/mol)	Polar Solvation Energy (kJ/mol)	SASA Energy (kJ/mol)	Binding Energy (kJ/mol)
α-Glucosidase-isoeugenol	−224.811	−10.382	83.618	−26.746	−186.222
α-Glucosidase-acarbose	−134.192	−4.813	62.125	−9.310	−90.102
α-Amylase-isoeugenol	−218.568	−29.891	62.172	−21.886	−180.194
α-Amylase-acarbose	−130.161	−2.106	39.340	−9.564	−87.109
HSA-isoeugenol	−189.601	−15.288	56.9638	−9.149	−98.169
HSA-aminoguanidine	−150.719	−5.127	47.498	−7.981	−81.872
HAR-isoeugenol	−184.951	−10.105	87.107	−25.191	−99.171
HAR-quercetin	−171.669	−3.291	81.102	−8.781	−85.768

**Table 9 molecules-27-06222-t009:** Druglikeness and pharmacokinetic properties of the representative compounds.

Properties	Isoeugenol	Acarbose	Aminoguanidine	Quercetin
Oral bioavailability (OB ≥ 30%)	Pass	Fail	Pass	Fail
Blood–brain barrier (BBB)	Moderate	Fail	Moderate	Pass
Drug half-life (HL < 3 h)	0.880	0.546	0.714	0.929
Lipinski’s rule (LR) of five	Accepted	Rejected	Accepted	Accepted
Intestinal epithelial permeability (Caco-2 cells)	−4.579	−6.149	5.448	−5.204
Drug-induced liver injury (DILI)	Negative	Positive	Negative	Positive
Clearness (CL > 15 mL/min/kg)	13.435	0.373	5.857	8.284
Molecular weight (MW 100~600)	164.080	645.250	74.060	302.040
Hydrogen bond acceptor (0~12)	2	19	4	7
Hydrogen bond donor (0~7)	1	14	6	8
TPSA (0~140)	29.460	321.170	87.920	131.360
PAINS	0	0	0	1

**Table 10 molecules-27-06222-t010:** PASS analysis of the representative compounds.

Compounds	α-Glucosidase		α-Amylase		AGE-Related Disorder		HAR	
	Pa	Pi	Pa	Pi	Pa	Pi	Pa	Pi
Isoeugenol	0.107	0.025	0.184	0.065	0.371	0.011	0.064	0.041
Acarbose	0.451	0.009	0.943	0.000	-	-	-	-
Aminoguanidine	-	-	-	-	-	-	-	-
Quercetin	0.139	0.058	-	-	-	-	0.466	0.003

## Data Availability

Not applicable.

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
