# Peer review of "Phyto-Computational Intervention of Diabetes Mellitus at Multiple Stages Using Isoeugenol from Ocimum tenuiflorum: A Combination of Pharmacokinetics and Molecular Modelling Approaches"

_molecules, 2022, doi:10.3390/molecules27196222_

Round 1
Reviewer 1 Report
In the present study, the authors report an in silico investigation of the antidiabetic potential of Ocimum tenuiflorum phytochemicals using computational methods including molecular docking, molecular dynamics simulations, and binding free energy calculations. ADMET, Druglikeness, Pharmacokinetics and PASS analyzes were also performed. Among the compounds examined, isoeugenol has been proposed as the best candidate acting as a multi-target inhibitor against all target enzymes. The in silico results are informative, valuable, and conclusive enough to indicate that their methodology could provide potential antidiabetic inhibitors. For this reason, the presented data take a step forward in the field of medicinal chemistry. Overall, the paper is clear, well-written and well-structured and can be published after considering the following points:
- The docking protocol must be validated.
- Gromacs software reference must be added.
- The software used for the visualization of the MD simulations must be mentioned
- I didn't understand why the authors chose isoeugenol as the best candidate. It can be observed from the data in Table 3 that some substances have higher affinities than isoeugenol. Please provide clarification.
- It is also reported in Table 3 that isoeugenol shows no hydrogen bonding with any proteins. Whereas in figure 1-4 we see the appearance of hydrogen bonds. Please provide explanations for this as well.
Reviewer 2 Report
It gives me pleasure to accept this invitation to review this manuscript. The manuscript entitled (Phyto-computational intervention of diabetes mellitus at multiple stages using isoeugenol from Ocimum tenuiflorum: A combination of pharmacokinetics and molecular modelling approaches) was well written. The authors do valuable work.
Page 2 line 75 Start with Ocimum tenuiflorum instead of O. tenuiflorum
Page 4 line 175 (supplementary Table S1) I could not find any supplementary file to download
Page 5 and 6 table 2: Adjust the font size to be smaller (Negative, accepted, moderate head of columns 3, 7, 8, 9, 10, 11, 12)
Page 6 table 3: Adjust the font size to be smaller head of all columns
Page 13 caption of figure 5 line 2: α-glucosidae not α- glucosidae
Page 15 caption of figure 6 line 2: α-amylase not α - amylase (2 times)
Page 19 Table 8: row 4 α-amylase-isoeugenol not α-amylase- isoeugenol
Row 6 HAS-isoeugenol not HAS- isoeugenol
Row 8 HAR-isoeugenol not HAR- isoeugenol
-Conclusion: Add recommendation and remarks of this computational study. (Start in vitro and in vivo studies of isoeugenol then clinical trials as antidiabetic).
